# Assessing the impact of imperfect adherence to artemether-lumefantrine on malaria treatment outcomes using within-host modelling

Joseph D. Challenger[1], Katia Bruxvoort[2], Azra C. Ghani[1] & Lucy C. Okell [1]

Artemether-lumefantrine (AL) is the most widely-recommended treatment for uncomplicated *Plasmodium falciparum* malaria worldwide. Its safety and efficacy have been extensively demonstrated in clinical trials; however, its performance in routine health care settings, where adherence to drug treatment is unsupervised and therefore may be suboptimal, is less well characterised. Here we develop a within-host modelling framework for estimating the effects of sub-optimal adherence to AL treatment on clinical outcomes in malaria patients. Our model incorporates the data on the human immune response to the parasite, and AL's pharmacokinetic and pharmacodynamic properties. Utilising individual-level data of adherence to AL in 482 Tanzanian patients as input for our model predicted higher rates of treatment failure than were obtained when adherence was optimal (9% compared to 4%). Our model estimates that the impact of imperfect adherence was worst in children, highlighting the importance of advice to caregivers.

[1] Medical Research Council Centre for Outbreak Analysis and Modelling, Department of Infectious Disease Epidemiology, Imperial College London, London W2 1PG, UK. [2] Department of Global Health and Development, London School of Hygiene and Tropical Medicine, London WC1E 7HT, UK. Correspondence and requests for materials should be addressed to J.D.C. (email: j.challenger@imperial.ac.uk)

The malaria parasite has thrived for millennia[1]. Despite concerted efforts, which have considerably reduced the burden of mortality and morbidity in recent years, in 2015 this parasite was still responsible for an estimated 212 million clinical cases and 429,000 deaths[2]. Artemisinin-based combination therapies (ACTs) are the first-line treatments for uncomplicated cases of *Plasmodium falciparum*[3]. The artemisinin derivative rapidly kills parasites, but is eliminated quickly by the body. Its partner drug has a longer half-life and clears parasites that remain after the artemisinin has been eliminated. ACTs have been instrumental in reducing malaria burden[4]. The majority of endemic areas have little or no drug resistance to ACTs and their high efficacy (~95%) at clearing parasitaemia has been extensively demonstrated in clinical trial settings[5,6]. However, there is less information on ACT effectiveness in routine health care settings when treatment is not supervised. Understanding how real-life patterns of patient adherence alter effectiveness, and how these can be improved is critical to maximising the role of ACTs in combating malaria.

Artemether-lumefantrine (AL) is the most widely administered antimalarial globally, being the first or second-line treatment in 30 countries in Africa, 10 in the Americas, 2 in the Eastern Mediterranean, 6 in South-East Asia, and 5 in the Western Pacific, as of 2015[2]. In 2013, it represented 73% of antimalarials procured worldwide[7]. The recommended course of treatment is 6 doses to be taken over 3 days, in contrast to the other ACTs currently recommended by the World Health Organisation which have only 3 doses taken over 3 days[3]. The increased number of doses has raised concerns about whether patient adherence to AL may be a problem in routine health settings.

Adherence to treatment can be assessed in a number of ways, including self-report by the patient or care giver, counting any pills remaining after treatment, measuring the timing of doses electronically or estimating doses taken by measuring drug concentrations present in the blood[8,9]. Measured patient adherence to AL varies from under 50 to 100%, dependent on setting and definition of adherence[9]. It is not easy to ascertain the effects of imperfect adherence on treatment outcomes. This can be studied indirectly by assessing the efficacy of supervised treatment compared with unsupervised treatment. Rahman et al. found that efficacy of treatment was high for both supervised and unsupervised patients in Bangladesh[10]. Studies of unsupervised usage of AL in Papua New Guinea have showed conflicting results[11,12]. In young Tanzanian children treated with AL, high cure rates were reported in supervised- and unsupervised-treatment groups; however, the median blood concentration of lumefantrine was lower in the unsupervised group, suggesting suboptimal adherence[13].

While these studies are informative, due to the complexity of adherence behaviour that varies not just by number of doses taken but also by timing, it is not possible from the above studies to identify which aspects of adherence are most important to ensure treatment success. The number of patients with any specific type of non-adherent treatment pattern in each study is small, and it is difficult to infer relationships from multiple studies due to varying degrees of immunity, which influences treatment success.

Given these challenges, there is scope for mathematical modelling work to complement the studies described above. A number of models combine a description of the within-host dynamics of parasitaemia with the pharmacokinetics (PK) and pharmacodynamics of antimalarial drugs[14–20]. Models describing within-host parasitaemia, are either simple and focus on the onset of a malaria infection[15,17–19], or more complicated and characterise the whole infection, including the immune response mounted against the parasite[19,20]. Typically, these models assume full adherence to treatment. A previous study modelled the effects of imperfect drug adherence on treatment failure, finding that treatment efficacy for AL was quite robust to skipping or delaying doses[15]. However, this model did not incorporate immune responses, which inform longer-term dynamics of recrudescent infections. The most detailed information on the parasite dynamics and the response of the human immune system is from a study of patients with neurosyphilis who were infected with *P. falciparum* as treatment for the bacterial infection. This "malaria therapy" data set provides an opportunity to study untreated infections of *P. falciparum* malaria[21], and has informed many within-host models[21–27]. Antigenic variation in the parasite population leads to complex within-host dynamics[28,29] and must be considered in models of the whole course of untreated *P. falciparum* infections.

Here we model the impact of imperfect patient adherence to AL using a multi-faceted approach. We develop a model of an untreated *P. falciparum* infection within the human host, simplifying parasite dynamics to a single equation, while keeping good fidelity to the data and still reproducing the effects of antigenic variation. We utilise an existing population PK model and calibrate a pharmacodynamic model to match observed recrudescence rates in clinical trials of AL, where treatment is supervised. We then model the effects of imperfect drug adherence on patients' clinical outcomes, utilising the data on patient adherence to AL collected in Tanzania. When we use these data

**Table 1 Comparing the model output with the malaria therapy data set**

|  | Minimum value | Median value | Maximum value |
|---|---|---|---|
| Initial slope: increase in $\log_{10}$ parasitaemia per day | *0.24* 0.19 **0.25** | *0.51* 0.49 **0.49** | *0.75* 0.87 **0.71** |
| Number of local peaks in parasitaemia | *2* 2 **1.7** | *9* 10 **7.7** | *17* 17 **15.5** |
| Slope of successive local peaks in parasitaemia | *−0.091* −0.074 **−0.03** | *−0.015* −0.013 **−0.010** | *−0.007* −0.001 **−0.002** |
| $\log_{10}$ of parasite density at 1st peak | *3.69* 3.37 **3.57** | *4.78* 4.79 **4.77** | *5.67* 5.66 **5.61** |
| Geometric mean of intervals between peaks in parasitaemia | *1.5* 14.4 **12.2** | *14.6* 20.0 **22.8** | *28.4* 77.8 **61.0** |
| Last day of patent parasitaemia | *38* 37 **39** | *193* 215 **198** | *404* 405 **409** |
| Proportion of parasite-positive observations in 1st half of infection | *0.57* 0.4 **0.46** | *0.97* 0.88 **0.97** | *1.0* 1.0 **1.0** |
| Proportion of parasite-positive observations in 2nd half of infection | *0.12* 0.08 **0.10** | *0.58* 0.46 **0.79** | *1.0* 1.0 **1.0** |
| Standard deviation of the log of the intervals between consecutive local peaks | *0.04* 0.03 **0.03** | *0.31* 0.2 **0.22** | *0.56* 0.47 **0.48** |

Here we display the values of the nine summary statistics describing dynamics of untreated infections in the malaria therapy data used to fit the model, as in ref. [30]. The central values are the data, values on the left (italics) are best-fit values from the model by Johnston et al., and values on the right (bold) are the best-fit values for our model (see Supplementary Methods for details and a description of the summary statistics). All quantities are based on parasitaemia detectable via microscopy. A local peak in the parasitaemia was defined as a parasite density greater than the three previous model observations, and greater than or equal to the next three observations. The first and second halves of the infection are defined via the first and last days that the patient was parasite positive by microscopy

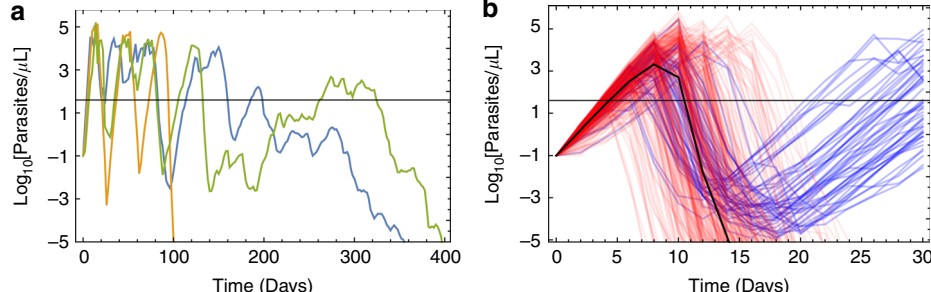

**Fig. 1** Simulated parasitaemia within individual patients. Here we show example model output for the parasite density (the number of parasitised red blood cells per μL) over time. Our within-host parasitaemia model was fitting to data from 35 malaria therapy patients, as described in the 'Methods' and Supplementary Methods section. **a** Untreated infections, where each different coloured line represents a different patient. These three model simulations were selected to demonstrate the variety of infection duration generated by the model. **b** Treated infections in 1000 simulated patients treated with six doses of AL, with the median parasitaemia at each two-day time point shown in black. Treatment fails in about 5% of cases (here coloured blue), causing patients to relapse

to determine the timings of doses in our simulation model, we obtain a model-estimated failure rate of 9%. If we assume perfect adherence here, the model-estimated failure rate is only 4%, which highlights the impact imperfect adherence to treatment can have on patient outcomes.

## Results

**Dynamics of untreated *Plasmodium falciparum* infections**. First, we developed a stochastic model to capture the dynamics of an untreated *P. falciparum* infection. Including long-term as well as short-term dynamics of parasitaemia is important to inform realistic predictions about recrudescent infections. Our starting point was a model by Johnston et al.[30], which built on existing work by Molineaux et al.[21]. Following their approach, we model three immune responses mounted by the host: the innate (fever) response, a variant-specific antibody-mediated response and a general antibody-mediated response. However, in contrast to previous work, we adapted the model to describe total asexual parasitaemia rather than separate subpopulations of *var* variants, resulting in a more computationally efficient model with one equation rather than 50. The simpler framework presented two challenges: how to describe the fluctuating growth rates caused by antigenic switching, and how to capture the interactions between the immune response and changes in parasite *var* presentation. To calibrate our model, we followed the approach of Johnston et al. who fitted to the minimum, median and maximum values of nine summary statistics, which describe the dynamics of infections in the malaria therapy data (Table 1), such as the duration of observed infection, and the rate of increase in parasite density at the beginning of the infection (see 'Methods' section).

The observed variation in parasite growth rate over the course of the infection (modelled as a variant-specific trait in ref. [25]) could be reproduced using a growth rate which varies stochastically over time but is correlated with the growth rate in the previous time point. Variant-specific immunity was approximated by an immune response which had limited memory of past parasitaemia, to mimic the parasite antigenic variation (see 'Methods' section). The structural form of our final model is displayed in Eq. 1 (see 'Methods' section) and Fig. 1a shows simulated parasitaemia over time in three example patients from the best-fit model (Supplementary Table 1). In general, we obtain a good fit to the target summary statistics (Table 1), with our new model describing aspects of both the initial acute phase of infection and later stages of infection well, including the first wave of parasitaemia, the decline in successive peaks of parasite density the duration of infection and the variation in these between patients. The model overestimates the proportion of

observations where parasitaemia was above the microscopy threshold during the later stages of infection, although more complicated models have also struggled to capture this[24].

**Population PK-pharmacodynamic model**. We modelled the impact of AL on parasitaemia in each patient, simulating blood drug concentrations using a published PK model[26], and estimating pharmacodynamic parameters using the clinical trial data. The published PK model was previously fitted to the data from adults and children in Tanzania treated with AL for clinical episodes of uncomplicated falciparum malaria[31] and includes both the effect of body weight and population-level variation in PKs (Supplementary Fig. 2). This model also describes the concentrations of the active metabolites of both artemether and lumefantrine, dihydroartemisinin (DHA) and desbutyl-lumefantrine (DLF), respectively. We modelled the standard course of AL, which requires twice-daily doses for 3 days (see 'Methods' section for recommended dose timings).

Time between infection and treatment was modelled by using a previously-defined pyrogenic threshold (the parasite density at which fever commences) which varies significantly between patients[32], and combining this with a distribution of times subsequently taken to seek treatment using self-reported times obtained from Demographic and Health Surveys[33] in sub-Saharan Africa (see Supplementary Methods).

Whereas the PK properties of the antimalarial drugs can be measured directly through the concentrations of the drugs in the blood, the pharmacodynamic properties of the drugs must be inferred from their effects on the parasitaemia. To do this, we matched treatment outcomes to the data from a clinical trial in which AL was used to treat uncomplicated falciparum malaria in children in Ndola, Zambia. The low prevalence of falciparum malaria in this region at the time of the study[4] suggests that these children should have low levels of naturally acquired immunity. In the trial, all doses were administered under direct supervision, to ensure adherence to treatment was optimal. We simulated a cohort of patients with a body weight distribution consistent with the study population[6]. We fit our PD model (Eq. 7) to two metrics: the proportion of parasite-positive patients at day 1 (i.e., 24 h after the first dose was administered), and the PCR-corrected recrudescence rates at day 28 (see 'Methods' section). Both these outcomes were required to determine the best-fit PD parameters, which are given in the 'Methods' section. Figure 1b illustrates the impact of treatment in our simulations, including some cases where the antimalarials do not clear the parasite population. For the body-weight distribution used to fit the

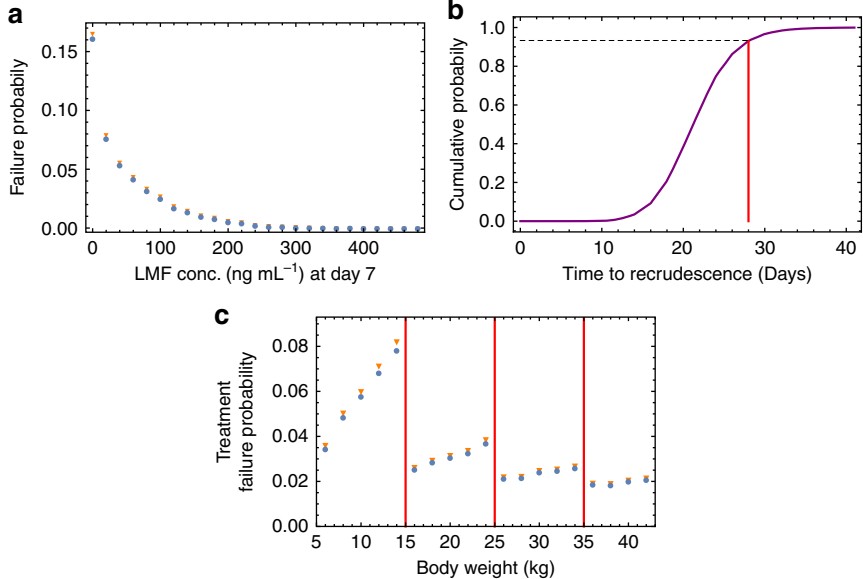

**Fig. 2** Details of treatment failure in our within-host model. Here we summarise simulated data generated from the proportion (about 5%) of $10^6$ model runs for which treatment failed. **a** The estimated probability of parasitological treatment failure according to the lumefantrine concentration 7 days after treatment commenced. The proportion failing treatment was calculated from $10^6$ model runs, grouping the results into bins of width 20 ng mL$^{-1}$. The blue circles measure treatment failure as positive by microscopy on day 28: the orange triangles show results obtained when an additional follow-up is made on day 42. **b** The simulated cumulative probability distribution for the time to recrudesce (the time at which parasitaemia becomes detectable by microscopy after treatment failure) in our model. By day 28 (red vertical line), around 93% (black dashed line) of recrudescences are detectable. **c** Factors influencing treatment failure. The probability of treatment failure according to patient body weight. The vertical red lines indicate the four dosing groups, determining the number of pills per dose (one, two, three, or four). These results were generated by grouping patients into 2 kg-wide bins. The symbols are as described in **a**

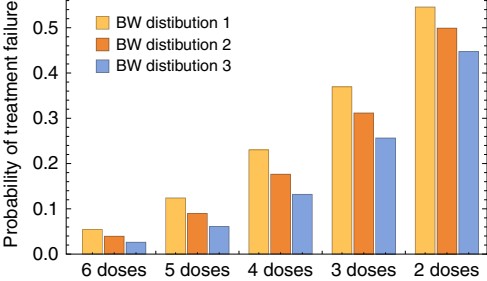

**Fig. 3** Model-estimated parasitological treatment failure rates at 28 days after treatment with AL. Results for perfectly-adherent patients (6 doses) are compared with patients who did not complete treatment. The bar marked '5 doses' denotes that only the first 5 doses were taken, '4 doses' that only the first 4 were taken, and so on. In each case we performed simulations for three body-weight distributions. For each bar shown, results were obtained by simulating a cohort of $10^5$ patients. Body-weight distribution 1, taken from clinical trials of AL in African children[6], was the distribution used when fitting the PD model (mean = 11.1 kg, SD 2.8). We compare the results obtained in that case with two older cohorts: distribution 2 (mean = 17.3 kg, SD = 6), and distribution 3 (mean = 30.0 kg, SD = 10.0). All distributions were Gaussian, truncated so that the minimum body weight was 5 kg

model, about 5.7% of simulated patients failed treatment (defined as the presence of a microscopy-detectable infection at day 28).

**Factors influencing treatment failure.** Population-level variation in PKs leads to wide variation in drug concentrations, even when all doses are supervised (Supplementary Fig. 2). In our simulations of perfectly-adherent patients (i.e., all doses contained the correct number of pills, and were taken at precisely the

recommended timings), low blood concentration of lumefantrine on day 7 after treatment, particularly <100 ng/ml, is strongly associated with increased probability of treatment failure (Fig. 2a), here defined as being parasite-positive by microscopy on day 28. This matches well with previous clinical trial observations[34,35]. We estimate that in perfectly-adherent patients who fail treatment, the median time to becoming microscopy-positive is 21 days (Fig. 2b). We find that a follow-up at 28 days will detect the majority of treatment failure events (93%), and therefore use this treatment failure outcome for our analysis. However, in clinical trial settings, this could indicate that a later follow-up may detect additional recrudescences in a cohort of patients.

A patient's body weight affects the concentration of antimalarials in the blood and hence those at the low end of a dosing group have higher concentrations than at the high end. For example, by the dosing guidelines (see 'Methods' section), a patient weighing 14 kg will receive one tablet per dose, while a patient weighing 16 kg will receive two. In a simulated population of adults and children with varying body weight, we find variation in the probability of treatment failure with the patient's body weight (Fig. 2c). We estimate only a small effect in most body-weight categories, but a more substantial effect in the 5–15 kg category, causing an estimated ~4% difference in efficacy between the smallest and largest body weight. The average efficacy is predicted to improve overall with age due to the effect of body weight on PKs, even without acquired immunity. We repeated this analysis, including an additional follow-up at 42 days. These results are also shown in Fig. 2a, c.

**The impact of imperfect drug adherence on treatment outcomes.** We used the calibrated model to estimate the consequences of imperfect adherence to treatment. One reason for poor adherence may be that, if symptoms are alleviated after the first few doses, the patient may not complete the treatment

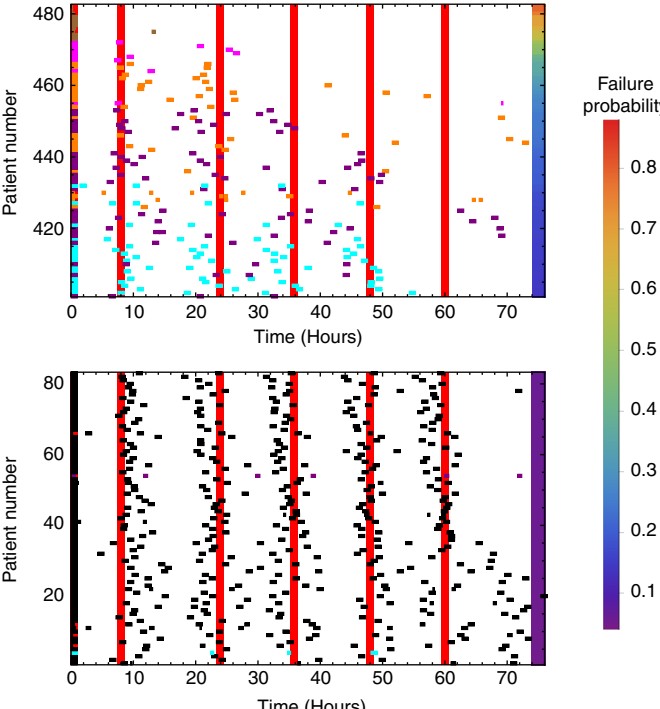

**Fig. 4** The time series data recorded by the smart blister packs in ref. [36]. and model-estimated probability of parasitological treatment failure at day 28. Each row represents a patient, with each dose taken marked by a rectangle. Shorter rectangles indicate that the patient did not take the full dose. The rectangles are coloured to indicate the percentage of prescribed pills taken by the patient: all pills (black) between 80–99% (cyan), 60–79% (purple), 40–59% (orange), 20–39% (magenta), and <20% (brown). The vertical, red lines indicate the recommended timings of the six doses. For patients who took multiple pills per dose, we have grouped pills into doses if pills were taken within half an hour of each other. In this figure, patients are ordered by the probability that their adherence profile results in treatment failure, according to our within-host model. As the full cohort is large (482 patients), we show two subsets here: the 80 patients with the lowest failure probability (lower panel) and the 80 with the highest failure probability (upper panel). The probability of failing treatment was estimated from $10^4$ simulations of the within-host model for each patient adherence profile. The body-weight data, which is needed to inform the PK model, were not available for this cohort, however, the dosing group (the number of pills per dose) is known. Therefore, when running our simulation model for each patient, we average over the weight range (e.g. 5–15 kg for the 1 pill per dose group) using a uniform probability distribution. Doses taken after 75 h are not shown here, but their effects are included in the model. The plot for all 482 patients is shown in Supplementary Fig. 4

regimen. To examine this, we simulated removal of doses one by one, starting with the 6th dose, then removing the 5th and 6th, and so on, assuming that the remaining doses were taken at the recommended time. These simulations were carried out using three different distributions for the patients' body weight. The estimated treatment failure rates clearly increase as doses are removed (Fig. 3), for example, from 5.7% when 6 doses are taken in the lowest body-weight group to 23% when only 4 doses are taken. The missing doses always caused highest predicted failure rates in the lowest body-weight group (mean = 11.1 kg, standard deviation = 2.8 kg), for example 23% vs. 13% in the largest body-weight group (mean = 30.0 kg, SD = 10.0) when 4 doses were taken.

We examined the relative importance of doses in relation to treatment failure in patients who took five timely doses, varying which dose was missed. For example, missing the fifth dose here means taking the first four doses at the recommended times, then taking a further dose at the recommended time for the sixth dose. Our results suggest that there was not a large difference in treatment failure probability dependent on which dose was missed. Missing the last dose was associated with a slightly higher probability of failing treatment (0.12 vs. 0.10 for the fifth dose, 0.093 for the fourth dose, 0.092 for the third dose and 0.088 for the second dose). These results were obtained with the body-weight distribution used to fit the model (mean = 11.1 kg, standard deviation = 2.8 kg).

We also assessed the importance of the timing of the 2nd dose, since the recommended time interval between the first and second doses is only 8 h and therefore when a patient commences treatment in the evening, the second dose will fall inconveniently in the middle of the night. We found that delaying the second dose by 4, 8, or 10 h did not lead to a raised probability of parasitological treatment failure at 28 days, even if the subsequent doses were also delayed. However, our model does not include important relevant outcomes such as alleviation of symptoms and prevention of hyperparasitaemia early in infection, which would potentially be affected by such a delay.

To simulate realistic adherence patterns, we also utilised the data collected in Tanzania in 2012 by Bruxvoort and colleagues, who monitored patients taking AL treatment, in which tablets were dispensed in smart blister packs. Microchips recorded the time at which pills were removed for 659 patients (see ref. [36]. for details of the data collection and ethical approval for the original study). These patients were largely unsupervised, although a minority of patients reported taking their first dose at the outlet where treatment was obtained[36]. According to this data, 67% of patients completed treatment, while 24% of patients took the correct number of pills for each dose at the correct time (allowing a tolerance of plus or minus four hours)[36]. We used these timings as input for our simulation model, to determine when each pill was taken. According to the data, 177 patients removed more pills from the blister pack than was stipulated in a single dose. We excluded these patients in our analysis, as our model does not consider the ill-effects of taking large doses of pills (e.g., toxicity). Equally, we do not know whether all the pills were taken at that time or just removed from the blister pack for later use. We therefore used the dose timings of 482 patients to estimate the probability of treatment failure at 28 days associated with each adherence profile (doses taken and timings). Figure 4 shows two subsets of the cohort with the lowest and highest estimated probability of treatment failure, as predicted by the model (Supplementary Fig. 4 displays results for all patients). Assessing the cohort as a whole, we estimated a failure rate at 28 days of 9% in these 482 patients. In comparison, a perfectly-adherent cohort with this body-weight distribution (see caption of Fig. 4 for details) has a failure rate of 4%. This 5% difference can be viewed as a penalty due to imperfect adherence to treatment.

We looked for particular patterns of drug adherence that were associated with an increased predicted risk in treatment failure and tested a number of measures (Table 2). When analysing all 482 patients, including the number of doses taken and the percentage of pills taken in the regression model explained much of the variation in the failure rates. As these two variables were strongly correlated, we included only the latter in the regression model. However, variables for all doses being timely and whether there was a long gap between doses did not improve the model fit and were not retained. We further assessed the impact of timely dosing in the subset of patients who took all pills prescribed in the correct number of doses. Including a variable for the sum of the differences (in hours) between the recommended and actual dose time for each patient indicated that, taking doses early was

**Table 2 Regression analysis of model results**

| Variable | Odds ratio | 95% confidence interval | *p*-value |
|---|---|---|---|
| Population model (n = 482) | | | |
| Proportion of prescribed pills taken | 0.00971 | (0.00738,0.0128) | <0.0001 |
| Subset Model (n = 305) | | | |
| Sum of [Actual minus recommended dose time] (in hours) | 1.009 | (1.007,1.012) | <0.0001 |
| All Doses Timely | 0.827 | (0.731,0.936) | 0.003 |

Predictors of treatment failure (measured at day 28) in our simulation model, based on patterns of adherence found in the individual-level data. Regression was performed using logit-transformed predicted treatment failure probability as the outcome. In model A all 482 patients were considered. A long gap between doses (>24 h) and all doses being timely (within 4 h of the recommended time) did not improve the model fit and were discarded. In Model B, we only assessed patients who had taken all prescribed pills in six doses. The categorical variable for a long gap between doses did not improve the model fit

predicted to give lower treatment failure probabilities than when doses were taken later. All doses being timely was also associated with lower treatment failure probability, whereas a categorical variable for having a long gap between doses did not improve the model fit.

## Discussion

In this work, we have combined a within-host model of a *P. falciparum* infection with the individual-level data on patient adherence to AL in Tanzania to estimate the impact of sub-optimal adherence on treatment outcomes. Importantly, the adherence data were collected from patients in routine health care settings. Using our within-host model, we estimate a treatment failure rate of 9% in this cohort of Tanzanian patients, which is higher than the failure rate of 4% predicted for this cohort had adherence to treatment been optimal. In future work, we aim to assess the impact that the increase in treatment failure rates has on onward transmission and the return of clinical symptoms.

Our results emphasise the importance of considering the large variation between patients in parasite dynamics and PKs. As known from clinical trials, full adherence to AL still leads to treatment failure in ~5% of patients. Our model matches well to the clinical trial data, indicating that low drug concentrations are a predictor of treatment failure. In a large pooled analysis of data from patients treated with AL, recrudescence was associated with low day-7 LMF concentrations and observed drug concentrations were lowest in very young children[35]. Conversely, in other patients, our results suggest that perfect adherence is not always required to successfully clear parasites. Using a model enabled us to quantify which aspects of adherence may be most important. Our results suggest that missing doses leads to higher percentage increases in treatment failure in younger patients, which underlines the fact that advice given to caregivers is very important.

As for other antimalarials, measures of adherence to AL show widely-varying results. In a recent review of studies of adherence to antimalarials[9], levels of adherence to AL varied from 38.7 to 100%, reflecting variation in patient adherence, but also differing definitions of being adherent to treatment. Clinical studies where treatment of AL was unsupervised show a range of results, with treatment failure ranging from rates similar to those observed in trials (~5% e.g., refs. [10,13]) to much higher levels (up to 20% e.g., refs. [11,37]). Our model-predicted failure rate of 9% in the Tanzanian cohort therefore falls within a plausible range. The variability in outcomes in these studies probably reflects differences in levels of adherence to treatment, although this was not assessed in the studies. Variation in immunity is also likely to have affected these outcomes. Two factors not included in our model due to lack of data that may explain imperfect adherence are early abatement of clinical symptoms and adverse side effects to the drugs. Unless these are significantly associated with parasitaemia and immunity (this may be the case for the former, but

we do not expect it to be so for the latter), they should not affect our conclusions. After ACT treatment of uncomplicated malaria, symptom resolution is rapid (typically about 1 day, see e.g., ref. [38]), and precedes complete clearance of parasites.

The modelling framework we have developed here could also be applied to other antimalarial drugs or candidate compounds in development (when the pharmacodynamics are characterised). The first step in the project was to develop a simple model of an untreated infection episode of falciparum malaria. While a number of similar models exist, many of these have a high level of complexity[21,24,25,30] and, as a result, are more computationally expensive. This is largely due to the details of the antigenic variation employed by the parasite, which presents a moving target to the human immune system, prolonging the duration of the infection. Here we were able to reduce the complexity to a one-equation model that is informed by the effects of the antigenic variation without modelling it explicitly, retaining good fidelity to the malaria therapy data set. This simplicity improves computational efficiency and hence has wider potential for integration into other frameworks.

It is important to note that the majority of malaria therapy patients, against whom we calibrated our model, had had no prior exposure to the *P. falciparum* parasite. This means that their adaptive immune response was generated solely in response to the malaria infection recorded in the dataset. Our model is therefore most relevant to individuals with little previous exposure, such as in low transmission settings. Other limitations of the model are that we did not include clinical treatment failure nor progression to severe malaria as outcomes, due to uncertainty over how and when these outcomes are triggered. We also did not include drug toxicity, which could arise, for example, when doses are taken too close together. Therefore, the model results suggesting that earlier dosing may reduce treatment failure rates could be misleading. Other issues also affect adherence to treatment such as drug side effects and vomiting a dose. Our model was fitted to match treatment outcomes found for a trial in which AL was given with fatty foods, which increases the bioavailability of lumefantrine[39]. Here we did not include the possibility that patients did not take the drug with food, which could provide an avenue for future work.

One modelling choice we had to make, for both treated and untreated infections, was to decide at what density of parasitaemia indicates that the malaria infection has been cleared. As mentioned in the 'Methods' section, we chose the density $10^{-5}$ PRBCs $\mu L^{-1}$, in line with other published within-host models. Varying this cut off will influence the proportion of infections that will be cleared by a course of AL. With the current cutoff we observe a failure rate of 5.7% in the cohort used to fit the model while, for example, a cutoff of $10^{-4.5}$ PRBCs $\mu L^{-1}$ results in a rate 4.2%, and a cutoff of $10^{-5.5}$ PRBCs $\mu L^{-1}$ leads to a rate 7.8%. Due to the inability to probe such low parasite densities in patients' blood and to assess the numbers of

sequestered parasites, it is unclear what the appropriate cut off should be, and indeed whether it is most appropriate to base it on total parasite numbers in a person, or a specific concentration in the blood. Longitudinal studies of patients using sensitive molecular methods such as qPCR would help to define if there is a low parasite concentration after which the parasite population is always cleared.

The advantage of a many-dose antimalarial regimen— allowing drug concentration to remain at effective levels for a long period of time— should be offset against the fact that a regimen with extra doses may lead to poorer adherence in unsupervised treatment settings. The effectiveness of unsupervised treatment as well as efficacy in clinical trials must be considered when comparing antimalarial treatments with differing dose regimens. All efforts must be made to ensure that patients understand the details of treatment regimens and the importance of completing treatment, both for ensuring optimal clinical outcomes for the individual, as well as minimising the duration of parasitaemia and onward transmission of the parasite.

## Methods

**Within-host model of an untreated infection.** We build upon a model originally proposed by Molineaux et al.[21] and extended by Johnston et al.[30] An important aspect of these models is the inclusion of the antigenic variation displayed by the parasite during an infection. A key target for the immune response is the protein PfEMP1, encoded by the *P. falciparum var* genes and expressed on the surface of a parasitised red blood cell (PRBC)[28,29]. This protein helps the parasites adhere to blood vessels to avoid clearance by the spleen. A small percentage of parasites switch their *var* expression each generation, presenting a moving target for the human immune system and lengthening the duration of an untreated infection. The published model describes the time evolution of 50 *var* variants during a single *P. falciparum* infection. Using the subscript $i$ to denote the variant being expressed, the model is written in terms of the number of parasitised red blood cells (PRBCs) per μL, $P_i(t)$:

$$P_i(t+1) = m_i[S_c(t)S_i(t)S_m(t)]P_i(t), \ i = 1, 2, \ldots, 50 \quad (1)$$

Here $m_i$ is the (constant) growth rate of variant $i$. This discrete-time model has a time-step of 48 h, reflecting the parasite's erythrocytic life span. Over the 48-h cycle, the PRBC spends part of its time in the blood stream, and the rest of the time adhering to blood vessel walls. Observation of a patient's blood during this period can show variation in the parasite density measured, depending on the proportion of parasites in the blood at a given time i.e. how synchronised the parasites' life cycles are[40]. One advantage of a 48-h time step is to discard this variation. In this published model, the immune response is described by three functions: the innate (or fever) response, $S_c(t)$; the adaptive response, divided into *var*-specific responses $S_i(t)$ (defined separately for each variant $i$); and a general adaptive response, $S_m(t)$, which acts against targets conserved across the variants. Total asexual parasite density is then defined as the sum of the parasite densities across all variants, $P_i(t)$. Variant diversity is generated by introducing a small probability of switching from variant $i$ to another variant at each time step. In the published model, the innate response depends simply on the current parasite density, whereas the adaptive immune functions depend on a weighted history of parasite density, designed to reflect a time-delayed adaptive immune response generating antibodies, which then subsequently wane over time.

Using this framework as a starting point, we sought to develop a model with only one equation for the total asexual parasitaemia:

$$P(t+1) = m(t)[S_c(t)S_v(t)S_m(t)]P(t), \quad (2)$$

where $m(t)$ represents the average growth rate of all the variants. Following the model outlined above, we define the beginning of the infection as $P(t=1) = 0.1$ PRBCs μL$^{-1}$ and deem an infection to have been cleared when parasitaemia falls below $10^{-5}$ PRBCs μL$^{-1}$ (about 50 parasites in total in the blood volume of an adult male). We retain the original form of the immune functions $S_c(t)$ and $S_m(t)$:

$$S_c(t) = \frac{1}{1 + \left(\frac{P(t)}{P_c^*}\right)^{\kappa_c}}, \quad (3)$$

$$S_m(t) = \frac{1-\beta}{1 + \left(\frac{\sum_{\tau=0}^{t-4} P_c(\tau)}{P_m^*}\right)^{\kappa_m}} + \beta, \ \text{where} \ P_c(\tau) = \begin{cases} P(\tau) \ \text{if} \ P(\tau) \leq C \\ C \ \text{if} \ P(\tau) > C \end{cases}. \quad (4)$$

However we replace the *var*-specific immunity functions $S_i(t)$ with an immune response $S_v(t)$ that we term the effective *var*-specific response (EVSR). This is equivalent to describing the average efficacy of the *var*-specific immune responses

at a particular time. The efficacy of this immune response depends on which variants are present at a given time and on the previous exposure of the immune system to these variants. Here we do this by giving $S_v(t)$ a memory of past parasitaemia, the duration of which will increase over the course of the infection. This effect is represented by the function $f(t)$, such that

$$S_v(t) = \frac{1}{1 + \left(\frac{\sum_{\tau=f(t)}^{t-4} P(\tau)}{P_v^*}\right)^{\kappa_v}}. \quad (5)$$

More detail on the motivation of the form of $S_v(t)$ can be found in the Supplementary Methods and we illustrate the EVSR in Supplementary Fig. 1.

**Capturing the growth rate of the parasite population.** In variant-specific models, the *var* variants grow at differing rates and the overall growth rate of the asexual parasitaemia is an average of the growth rates of the subpopulations present at any given time. In the data set, the log-transformed value of the initial slope of the parasite growth has a wide range (Table 1), reflecting a variety of growth rates early in the infections. Therefore, our effective growth rate, $m(t)$, should vary according to an appropriate probability distribution. However, it should also vary with time, as variants emerge and are subsequently removed from the infection.

We found that the best results are obtained when the growth rate at any one time is drawn from a probability distribution, but also positively correlated with the growth rate in the previous time step. This makes sense intuitively: after entering the parasite population, a *var* typically remains present in the infection for a number of generations before being cleared, contributing to the average growth rate over this time. As the percentage of merozoites that switch their *var* expression per generation is low, the average growth rate at time $t$ should be positively correlated with the average growth rate at $t+1$. More formally, using angled brackets to denote the average or expected value, we write

$$m \sim N(\mu, \sigma), \ \frac{\langle (m(t) - \mu)(m(t+\theta) - \mu) \rangle}{\sigma^2} = g^\theta, \ 0 < g < 1. \quad (6)$$

Hence, the growth rates at time $t$ and $t+1$ are positively correlated with correlation g. As in refs. [21,30]. We truncate the distribution at 1 and 35, thus limiting the minimum and maximum possible growth rates of the variants.

The model was fitted to summary statistics from the malariatherapy data set (Table 1), using Markov chain Monte Carlo methods to generate random walks in parameter space. See Supplementary Methods for full details of model fitting. The parameters varied in the MCMC analysis were $\sigma, g, k_m, P_v^*$, as well as the parameter that characterises $f(t)$ in Eq. 3. The best fit parameters are shown in Supplementary Table 1. In order to fit the data, we had to strengthen the general-adaptive immune response, which means that the balance between the two adaptive responses has changed compared to the previous models.

**The pharmacokinetic and pharmacodynamic models.** We utilised a published pharmacokinetic model by Hodel et al., fitted to the data from patients in Tanzania treated with AL following clinical episodes of uncomplicated falciparum malaria[31]. Following the first dose, subsequent doses are administered at 8, 24, 36, 48, and 60 h. Each tablet contains 20 mg of artemether (AM) and 120 mg of lumefantrine (LMF), with the doses determined by the patient's weight: 5–15 kg: one tablet per dose; 15–25 kg: two tablets per dose; 25–35 kg: three tablets per dose; > 35 kg: four tablets per dose[3]. Supplementary Fig. 2 shows the blood concentrations of the drugs and their metabolites, including the population-level variation.

The pharmacodynamic component of the model determines the rate at which a drug kills parasites, as a function of its concentration. Following other analyses[19,20] we write this effect as

$$\frac{dP}{dt} = -k_{max}\left[\frac{C(t)}{C(t) + C_{50}}\right]P. \quad (7)$$

The drug effect saturates at high concentrations, with a maximum kill rate of $k_{max}$. The concentration at which the maximum drug effect is halved is denoted by $C_{50}$. In other modelling work a Hill-coefficient has been added to this equation e.g., ref. [16]. However, without detailed time-series information on the effects the drugs have on the parasitaemia, it is difficult to determine this coefficient. Therefore, here we set it equal to one.

Since we have two drugs and two metabolites, we could in theory attempt to fit these two parameters for all of quantities, which would mean eight PD parameters in total. However, with the data at hand, it would be impossible to uniquely determine all these parameters. Therefore, we take steps to reduce the parameter space, firstly by giving AM and DHA common values for the parameters $k_{max}$ and $C_{50}$. Also, it is clear from the PK modelling that the concentration of metabolite — DLF (Supplementary Fig. 2) is much lower than that of LMF. We will assume that the dominant effect comes from the LMF, and ignore the PD properties of DLF[41], meaning that the contribution of the partner drug is only represented through the concentration of LMF. This may have the effect of slightly overestimating the killing rate of LMF, but we do not expect this effect to be very large.

From a large, multi-centred study of ACTs in African children[6], we selected clinical trial results from Ndola in Zambia against which to calibrate our pharmacodynamic model. This site was chosen due to the low malaria prevalence observed over the timespan of the trial[4]. As our model of the parasitaemia is fitted

to the data in patients with no prior exposure to the parasite, we expect our model to be most relevant in low transmission settings. At this site, 71 out of 75 patients were parasite negative 28 days after treatment commenced, once new infections had been accounted for by PCR[6]. This gives a treatment success rate of nearly 95%. This is consistent with AL success rates observed in the other trial sites, as well as in other studies[5,6]. Our PD model, then, should be consistent with this success rate. However, we found that this alone was not sufficient to constrain the PD parameters. We augmented this information with the percentage of patients who were parasite positive one day after treatment commenced. Using these two statistics, we simulated cohorts of 75 patients with a body-weight distribution to match the Ndola cohort (normally distributed with mean and standard deviation of 11.1 and 2.8 kg, respectively). Simulating a large ensemble of cohorts generates a distribution of values for the two target statistics, due the stochastic nature of the model. The likelihood cannot be calculated explicitly from our stochastic simulation model. However, we constructed a so-called pseudo likelihood, using repeated stochastic simulation for each set of model parameters[42]. In addition, we used estimates of the parasite reduction ratio[43], along with in vitro estimates of the drugs' properties[44–46] to constrain the parameters. The results from the studies on recrudescence[35] and reinfection[47] following treatment with AL were led us to require $C_{50} > 100$ ng mL$^{-1}$ for LMF. The best-fit PD parameters for LMF were $k_{max} = 0.165$ h$^{-1}$ and $C_{50} = 125$ ng mL$^{-1}$, and for artemether (and metabolite DHA) $k_{max} = 0.189$ h$^{-1}$ and $C_{50} = 3.3$ ng mL$^{-1}$. Details of how the parasitaemia, PK, and PD models were combined can be found in the Supplementary Methods and in Supplementary Fig. 3.

**Ethics statement.** All the data analysis conducted during this research was secondary and used studies that had obtained ethical approval previously from the appropriate organizations. All the data were anonymised before being provided to investigators.

**Data availability.** The source code for the within-host model can be found at: http://github.com/JDChallenger/adherence_project. The individual-level adherence data for the Tanzanian cohort can be accessed through the ACTc Publications Datasets Repository at http://actc.lshtm.ac.uk.

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

## Acknowledgements

This work was funded by Medicines for Malaria Venture. L.C.O. also acknowledges funding from a UK Royal Society Dorothy Hodgkin Fellowship. We additionally acknowledge wider support from the Bill and Melinda Gates Foundation and the UK Medical Research Council. With permission, we have used the IMPACT2 Adherence Study Data. IMPACT2 was funded by the ACT Consortium, through a grant from the Bill and Melinda Gates Foundation to the London School of Hygiene and Tropical Medicine. We acknowledge the IMPACT2 staff: Charles Festo and Admirabilis Kalolella (both Ifakara Health Institute), Katia Bruxvoort, Catherine Goodman and David Schellenberg (all London School of Hygiene and Tropical Medicine).

## Author contributions

A.C.G. and L.C.O. conceived the project; J.D.C. developed the model and wrote the first draft of the manuscript; J.D.C. analysed the data with contribution from K.B. All authors edited and commented on the manuscript.

## Additional information

**Competing interests:** The authors declare no competing financial interests.

