## [Peer Review File · Nature Communications]

Reviewers' comments:

Reviewer #1 (Remarks to the Author):

In the manuscript entitled, "Assessing the impact of imperfect adherence to artemether-lumefantrine on malaria treatment outcomes using within-host modelling", the authors (Challenger et al.) present the results of an ambitious project to quantify the efficacy gap between treatment outcomes for a frontline anti-malarial drug under supervised (i.e., clinical trial) versus non-supervised (i.e., 'real world') settings. The results of this study will be of interest to governmental and NGO workers engaged in shaping public health policy within endemic countries and to the academic modelling community of specialists in malaria and global burden of disease studies. The methodology adopted comprises three stages of model fitting to link stochastic models for the course of infection and host immune response to the PK/PD action of the two compound therapy. While overall I would commend this manuscript as being of Nature Communications quality, I raise below a number of concerns I would hope to see addressed.

First of all, I am concerned that the end point of the modelling here---namely, 'treatment failure' defined as a patient remaining parasite positive by microscopy at day 28 after treatment---is perhaps shaped more by observational pragmatism rather than epidemiological significance. That is, while this definition may be convenient in-field for cohort studies it is not clear how it might relate to the probability of onwards transmission, return of clinical symptoms, or emergence of drug resistance. To some extent this is an issue of presentation: making clearer the motivations of the study and interpretation of the results. However, I wonder whether some qualitative insight into these alternative understandings of treatment failure could be recovered from the existing model? Presumably the risk of return of clinical symptoms could be quantified from the pyrogenic threshold; (more ambitiously) perhaps also presence of non-trivial gametocytemia could also be inferred from its approximate relationship with lagged parasite density; similarly, could the 28 day end point be relaxed once the model is calibrated against the 28 day data?

On a related theme (use of the pyrogenic threshold), regarding the section of the results dealing with early stopping of treatment I wonder whether the possible causative action of treatment abandonment due to early abatement of symptoms could be represented in these simulations? A "Devil's Advocate" position would be to suggest that the cohort of early stoppers are much less likely to suffer recrudescence because they have the strongest immune responses and/or the greatest absorption of the drug.

Some technical questions regarding the model specification and fits:

- I am curious about a possible sensitivity of these results to the threshold chosen for declaring end of infection: the threshold of 10^{-5} PRBCs per microlitre is noted as representing about 50 parasites in the total blood volume of an adult male, but the target study cohort subsequently is that of young children for whom this must be closer to 10-20 parasites. Perhaps not a huge difference on paper but of possible interest to explain for those of us without firsthand experience with this particular model.

- Regarding validation of the single equation model as a sensible reduction of the 50 equation base model one could examine the similarities and differences between the fitted *non-adaptive* immune responses: if the new predicted dynamics were substantially different from the earlier publication one might caution against viewing the single equation model as a close proxy of the more complex base model.

- While I understand the challenges of identifiability with limited data, I wonder about the decision to drop DLF from the PD model since I've been sensitised to the notion that the long half life of the partner drug makes it the greatest killer of parasites in a typical combination therapy (e.g. Hasting, Hodel & Kay 2016; Nature Scientific Reports). Certainly it seems possible from Supp Fig 2

that the DLF half life is very much longer than that of Lumefantrine. Perhaps a worthwhile 'sanity check' would be to try dropping the Lumefantrine model in favour of DLF to assess impact.

- Regarding the results on probability of dosing failure with treatment rate, I wonder whether there are any supporting studies publishing on this in the recent literature. I'm not familiar with the results for AL but certainly age banding and optimal pill sizing has been considered extensively for primaquine (e.g. Leang et al. 2016).

- In fitting of the PD model it did not seem that the method was indeed the usual Approximate Bayesian Computation since although summary statistics were used based on observed and mock datasets the comparison was via a pseudo-likelihood function (i.e., more like Bayesian indirect inference; Drovandi et al. 2015)---perhaps what falls under the "external error model and informal likelihoods" section of the given reference (Hartig et al.)---rather than a metric distance (box 2 of Hartig et al.).

- In fitting to the malaria therapy data I was confused as to why 50 samples was chosen as the smallest unit for computing the min, median, and maximum summary statistics since the observed data had 35 patients: the min and maximum being strongly dependent on the drawn sample size (unlike the median). That aside, the fitting method described may also have a connection to Approximate Bayesian Computation if viewed as the 'maximum ABC likelihood' fit taken by associating the parameters mapping to the closest point in binding function space to the observed summary data as the desired estimator (under the binding function definition of Frazier, Martin, & Robert 2015). When optimising with MCMC it can be helpful to gradually raise the inverse temperature of the chain to concentrate on the posterior mode $L(\theta)^{\beta}$ with $\beta : 1 \rightarrow$ something large (e.g. 5) since ordinary MCMC is designed to draw asymptotically from the posterior mass rather than to locate the posterior mode.

Reviewer #2 (Remarks to the Author):

Poor adherence to antimalarial treatment regimens is complex, and provides a large and uneven array of drug exposure patterns. It is undeniably an important issue.

This report describes a well performed fusion of an established intrahost model of *P. falciparum* parasite dynamics, a less well established effect of immunity on therapeutic responses, and a particular population PK model derived from a study in Tanzania. The selection of the PK and PK-PD variables can be questioned, but there is no need to. This particular set of simulations provided that particular array of predicted outcomes. These outcomes are sensitive to the parameterization, and the input variables chosen may, or may not, provide useful predictions. So the exercise was good, but was it useful?

I struggled to find an operationally relevant message from this well conducted modelling exercise beyond the need to study adherence as best we can in the different contexts of antimalarial drug use. I think we knew that.

Reviewer 1

"In the manuscript entitled, "Assessing the impact of imperfect adherence to artemether-lumefantrine on malaria treatment outcomes using within-host modelling", the authors (Challenger et al.) present the results of an ambitious project to quantify the efficacy gap between treatment outcomes for a frontline anti-malarial drug under supervised (i.e., clinical trial) versus non-supervised (i.e., 'real world') settings. The results of this study will be of interest to governmental and NGO workers engaged in shaping public health policy within endemic countries and to the academic modelling community of specialists in malaria and global burden of disease studies. The methodology adopted comprises three stages of model fitting to link stochastic models for the course of infection and host immune response to the PK/PD action of the two compound therapy. While overall I would commend this manuscript as being of Nature Communications quality, I raise below a number of concerns I would hope to see addressed."

"First of all, I am concerned that the end point of the modelling here---namely, 'treatment failure' defined as a patient remaining parasite positive by microscopy at day 28 after treatment---is perhaps shaped more by observational pragmatism rather than epidemiological significance. That is, while this definition may be convenient in-field for cohort studies it is not clear how it might relate to the probability of onwards transmission, return of clinical symptoms, or emergence of drug resistance. To some extent this is an issue of presentation: making clearer the motivations of the study and interpretation of the results. However, I wonder whether some qualitative insight into these alternative understandings of treatment failure could be recovered from the existing model? Presumably the risk of return of clinical symptoms could be quantified from the pyrogenic threshold; (more ambitiously) perhaps also presence of non-trivial gametocytemia could also be inferred from its approximate relationship with lagged parasite density; similarly, could the 28 day end point be relaxed once the model is calibrated against the 28 day data?"

We thank the reviewer for their perceptive comments and questions. The reviewer is correct that there are a number of possible definitions for treatment failure in this context. Our choice—a patient's being parasite positive by microscopy at day 28 after treatment—is indeed motivated by procedure in clinical trial settings, to which we calibrated our data. Once the model was fitted, using this definition of treatment failure, we were able to check whether day 28 was a suitable time point to use. Simulating a large number of model runs and focussing on the cases where the drugs did not clear the parasitaemia, we noted the first day on which patients became parasite positive by microscopy. As shown in Fig 1B, we found that in 7% of cases, the day 28 check would not detect the recrudescence parasitaemia. Following the reviewer's question, we have repeated the analyses used to generate the panels in Figure 1 to include an additional follow-up at day 42. When stratified by body weight or day 7 lumefantrine concentration, the results are extremely similar to those obtained by only using a 28 day follow up. The fact that some treatment failures in our model simulations are not detectable by microscopy at day 28 has been underlined in the manuscript: in the Results section (first paragraph of subsection 'Factors influencing treatment failure') we write: "... in clinical trial settings, this could indicate that a later follow-up may detect additional recrudescences in a cohort of patients". In the same subsection we indicate that Figure 2 now shows results when an additional follow-up is made at day 42: "We repeated this analysis, including an additional follow-up at 42 days. These results are also shown in Figs 2A and 2C."

Regarding the contribution of these cases to onward transmission, we agree that this issue is extremely important. We are currently looking at adding gametocytes to the model, but we prefer to keep this project separate, as this is a significant additional model development and there is already

a lot of material to present in the current paper. Similarly, the emergence of *de novo* resistance and the propagation of existing resistant parasites are very interesting, but would be a very substantial project in their own right if we are to fully validate and parameterise such a model, and are beyond the scope of this analysis.

With regard to clinical symptoms, treatment failure in non-immune patients will sometimes lead to the return of clinical symptoms. We agree that it would be interesting to look at the impact that imperfect adherence has on additional fevers. To do this properly, one should assess whether the pyrogenic threshold for subsequent fevers can be assumed to be the same as for the first fever. Because this is unclear, in the Discussion we have added that impact of poor adherence on onward transmission and subsequent fevers is an important avenue for further work. At the end of the first paragraph we write: "In future work, we aim to assess the impact that the increase in treatment failure rates has on onward transmission and the return of clinical symptoms."

On a related theme (use of the pyrogenic threshold), regarding the section of the results dealing with early stopping of treatment I wonder whether the possible causative action of treatment abandonment due to early abatement of symptoms could be represented in these simulations? A "Devil's Advocate" position would be to suggest that the cohort of early stoppers are much less likely to suffer recrudescence because they have the strongest immune responses and/or the greatest absorption of the drug.

The reviewer makes a very important point about the link between the abatement of symptoms and a patient's decision not to complete the course of treatment prescribed to them. Unfortunately we don't have information on patients' parasitaemia for this cohort (either at baseline or follow up), which makes this issue difficult to assess here. However, fever abatement after ACTs is fast: typically about one day, see e.g. Fig. 5 in "In vivo efficacy of artemether-lumefantrine and artesunate-amodiaquine for the treatment of uncomplicated falciparum malaria in children: a multisite, open-label, two-cohort, clinical trial in Mozambique" (Nhama et al. Malaria Journal 13:309 [2014]). This is, obviously, shorter than both the timescale for complete parasite clearance (i.e. the end of the infection) and the timescale of the recrudescence dynamics. While swift abatement of symptoms could be correlated with, say, low baseline parasitaemia or an effective immune response, we do not believe it will automatically lead to clearance of all parasites if subsequent doses are not taken. Furthermore, we also hypothesise that side-effects to the drugs could be a more critical reason for discontinuing treatment, which should not be related to symptoms. We have added these points to the Discussion section (third paragraph) of the article: "Two factors not included in our model due to lack of data that may explain imperfect adherence are early abatement of clinical symptoms and adverse side effects to the drugs. Unless these are significantly associated with parasitaemia and immunity (this may be the case for the former, but we do not expect it to be so for the latter), they should not affect our conclusions. After ACT treatment of uncomplicated malaria, symptom resolution is rapid (typically about 1 day, see e.g. Ref. 39), and precedes complete clearance of parasites."

"I am curious about a possible sensitivity of these results to the threshold chosen for declaring end of infection: the threshold of 10^{-5} PRBCs per microlitre is noted as representing about 50 parasites in the total blood volume of an adult male, but the target study cohort subsequently is that of young children for whom this must be closer to 10-20 parasites. Perhaps not a huge difference on paper but of possible interest to explain for those of us without firsthand experience with this particular model."

The reviewer is interested in our choice of threshold for declaring that the parasite population had been cleared. We used the threshold of 10^{-5} parasites per microliter which had been set in other

model analyses. The main reason for this was that we wished to compare our model results those obtained using more complicated models which explicitly modelled antigenic variation. Therefore, we preferred not to vary too many elements of the model, as this would 'muddy' the comparison between the models. Furthermore, the inability to probe such low parasite densities *in vivo* led us to choose a very simple, fixed cut off. However, we agree that the choice of the threshold should be investigated and have performed an additional modelling exercise to address this: raising & lowering the fixed threshold. We have added some details of this to the Discussion (penultimate paragraph), along with the possibility of adding a weight-dependent threshold in future work. We write, "One modelling choice we had to make, for both treated and untreated infections, was to decide at what density of parasitaemia indicates that the malaria infection has been cleared. As mentioned in the Methods section, we chose the density 10^{-5} PRBCs μL^{-1} , in line with other published within-host models. Varying this cut off will influence the proportion of infections that will be cleared by a course of AL. With the current cut off we observe a failure rate of 5.7% in the cohort used to fit the model while, for example, a cut off of $10^{-4.5}$ PRBCs μL^{-1} results in a rate 4.2%, and a cut off of $10^{-5.5}$ PRBCs μL^{-1} leads to a rate of 7.8%. Due to the inability to probe such low parasite densities in patients' blood and to assess the numbers of sequestered parasites, it is unclear what the appropriate cut off should be, and indeed whether it is most appropriate to base it on total parasite numbers in a person, or a specific concentration in the blood. Longitudinal studies of patients using sensitive molecular methods such as qPCR would help to define if there is a low parasite concentration after which the parasite population is always cleared. "

*"Regarding validation of the single equation model as a sensible reduction of the 50 equation base model one could examine the similarities and differences between the fitted *non-adaptive* immune responses: if the new predicted dynamics were substantially different from the earlier publication one might caution against viewing the single equation model as a close proxy of the more complex base model."*

This is something we were conscious of during model fitting. We were able to retain both the form and parameter values of the non-adaptive (fever) immune response, which controls the height of the first peak of asexual parasitaemia. We retained the form of the general-adaptive immune response, but we did have to refit the parameter values, making it more effective than in the full 50-equation models. This suggests that our model is not a perfect proxy of the full model, however, *in vivo*, it would be very challenging to assess which facet of the adaptive immune response was responsible for killing the most parasites at a particular point of an infection. It is interesting that we achieve similarly good fits after altering the balance between the immune responses. In the Methods section (end of subsection "Capturing the growth rate of the parasite population") we now have written, "In order to fit the data we had to strengthen the general-adaptive immune response, which means that the balance between the two adaptive responses has changed compared to the previous models."

"While I understand the challenges of identifiability with limited data, I wonder about the decision to drop DLF from the PD model since I've been sensitised to the notion that the long half life of the partner drug makes it the greatest killer of parasites in a typical combination therapy (e.g. Hasting, Hodel & Kay 2016; Nature Scientific Reports). Certainly it seems possible from Supp Fig 2 that the DLF half life is very much longer than that of Lumefantrine. Perhaps a worthwhile 'sanity check' would be to try dropping the Lumefantrine model in favour of DLF to assess impact."

Following the reviewer's question about our decision not to include the effects of metabolite DLF in the PD model, - we have now further examined the impact of including DLF in the model, instead of lumefantrine, as suggested. Although metabolite DLF is present in the body for a long period of time,

the concentrations observed are typically two orders of magnitude lower than for lumefantrine. Retaining the PD parameters fitted for lumefantrine resulted in an extremely large increase in the rates of treatment failure observed (>70% of infections failed to clear). To obtain results consistent with failure rates observed in clinical trials, then, would require large changes to the values of parameters k_{max} and C_{50} , to unrealistic levels - similar to values characteristic of artemisinin derivatives, which have very different properties. Therefore, we believe that DLF effects are very small relative to lumefantrine. Including DLF would require reducing the lumefantrine action, rather than changing the balance between the artemether and partner drug. Regarding the half-lives, there is a lot of individual-level variation in the model but on average DLF will not have a longer half-life than lumefantrine for the cohort studied in Suppl. Fig 2. The DLF peaks slightly later, however, as lumefantrine is metabolised slowly. We have slightly changed the relevant part of the Methods section (in subsection "The pharmacokinetic and pharmacodynamic models") to emphasise this: "We assume that the dominant parasite-killing effect comes from the LMF, and ignore the PD properties of DLF."

"Regarding the results on probability of dosing failure with treatment rate, I wonder whether there are any supporting studies publishing on this in the recent literature. I'm not familiar with the results for AL but certainly age banding and optimal pill sizing has been considered extensively for primaquine (e.g. Leang et al. 2016)."

Thanks for this suggestion. We are not aware of an equivalent study for artemether-lumefantrine looking at optimal dosing, however, a recent review & meta-analysis by the WWARN group (WorldWide Antimalarial Resistance Network) looked for patient factors that influenced the Day 7 blood concentration of lumefantrine, which has previously been shown to be a good indicator of treatment efficacy (see BMC Medicine 13:227 [2015], doi: 10.1186/s12916-015-0456-7). Their findings support our conclusion that young children are at risk of under-dosing. We have made the connection with this paper clearer now in the second paragraph of the Discussion: "Our model matches well to clinical trial data indicating that low drug concentrations are a predictor of treatment failure. In a large pooled analysis of data from patients treated with AL, recrudescence was associated with low day 7 LMF concentrations and observed drug concentrations were lowest in very young children".

"In fitting of the PD model it did not seem that the method was indeed the usual Approximate Bayesian Computation since although summary statistics were used based on observed and mock datasets the comparison was via a pseudo-likelihood function (i.e., more like Bayesian indirect inference; Drovandi et al. 2015)---perhaps what falls under the "external error model and informal likelihoods" section of the given reference (Hartig et al.)---rather than a metric distance (box 2 of Hartig et al.)."

"In fitting to the malaria therapy data I was confused as to why 50 samples was chosen as the smallest unit for computing the min, median, and maximum summary statistics since the observed data had 35 patients: the min and maximum being strongly dependent on the drawn sample size (unlike the median). That aside, the fitting method described may also have a connection to Approximate Bayesian Computation if viewed as the 'maximum ABC likelihood' fit taken by associating the parameters mapping to the closest point in binding function space to the observed summary data as the desired estimator (under the binding function definition of Frazier, Martin, & Robert 2015). When optimising with MCMC it can be helpful to gradually raise the inverse temperature of the chain to concentrate on the posterior mode $L(\theta)^{\beta}$ with $\beta : 1 \rightarrow \text{something large (e.g. 5)}$ since ordinary MCMC is designed to draw asymptotically from the posterior mass rather than to locate the posterior mode."

The reviewer is right that our use of the term, 'Approximate Bayesian Computation' was not correct- we apologise for any confusion caused. We have amended the manuscript to clarify this. The

method is better described as constructing a pseudo likelihood (sometimes called synthetic likelihood, as described in Fig 5b of the paper by Hartig). We have amended the Methods section (in the final paragraph) as follows: "The likelihood cannot be calculated explicitly from our stochastic simulation model. However we constructed a so-called pseudo likelihood, using repeated stochastic simulation for each set of model parameters". We have made a similar correction in the Supplementary Methods (in the final paragraph).

The reviewer is right that, in fitting to the malaria therapy data, the number of samples chosen will influence the values minimum and maximum summary statistics. We followed the choice of Johnston et al. (PloS Comput. Biol. 9(4) e1003025 [2013]) as we wished to compare our results to theirs, as shown in Table 1.

Reviewer 2

Poor adherence to antimalarial treatment regimens is complex, and provides a large and uneven array of drug exposure patterns. It is undeniably an important issue.

*This report describes a well performed fusion of an established intrahost model of *P. falciparum* parasite dynamics, a less well established effect of immunity on therapeutic responses, and a particular population PK model derived from a study in Tanzania. The selection of the PK and PK-PD variables can be questioned, but there is no need to. This particular set of simulations provided that particular array of predicted outcomes. These outcomes are sensitive to the parameterization, and the input variables chosen may, or may not, provide useful predictions. So the exercise was good, but was it useful?*

I struggled to find an operationally relevant message from this well conducted modelling exercise beyond the need to study adherence as best we can in the different contexts of antimalarial drug use. I think we knew that.

Artemether-lumefantrine is the most widely-used treatment for uncomplicated falciparum malaria worldwide. The large majority of AL treatment courses are prescribed in routine health care settings, where patients are unsupervised except sometimes for the first dose. Collecting data to link particular types of poor adherence with treatment outcomes is extremely difficult, not least for ethical reasons. We believe, therefore, that estimates of efficacy from a within-host model, properly fitted to available data, is a useful contribution.

As for all such models, the results will be sensitive to the parameterisation used. In particular, the values for our PD parameters will depend on (i) The details of the asexual parasitaemia model (ii) The population PK model utilised (iii) The particular form of the pyrogenic threshold (iv) The choice of delay between the onset of clinical symptoms and treatment being obtained. For each of these, therefore, we have taken care to source relevant data to fit each facet of the model, before turning to fit the PD model.

We believe that our findings are operationally relevant. In particular, the finding that adherence in young children is especially important should be emphasised in communication to caregivers, as we stressed in the second paragraph of the Discussion. The relative robustness of AL to e.g. delaying individual doses is

notable, and is an advantage for this treatment regimen, although care should be taken in communicating this message such that it is not mistaken as saying that adherence is not important.

We thank both reviewers for their comments.

Joseph Challenger

(on behalf of all the authors)

REVIEWERS' COMMENTS:

Reviewer #1 (Remarks to the Author):

With their reply to the referees and additional test simulations the authors have thoroughly addressed my questions regarding the original analysis. I would be happy to recommend this article for publication.